# REPRESENT TO CONTROL PARTIALLY OBSERVED SYSTEMS: REPRESENTATION LEARNING WITH PROVABLE SAMPLE EFFICIENCY

**Lingxiao Wang** [*], **Qi Cai** [*]
Department of Industrial Engineering and Management Sciences
Northwestern University
{lingxiaowang2022,qicai2022}@northwestern.edu

**Zhuoran Yang**
Department of Statistics and Data Science
Yale University
zhuoran.yang@yale.edu

**Zhaoran Wang**
Department of Industrial Engineering and Management Sciences
Northwestern University
zhaoranwang@gmail.com

## ABSTRACT

Reinforcement learning in partially observed Markov decision processes (POMDPs) faces two challenges. (i) It often takes the full history to predict the future, which induces a sample complexity that scales exponentially with the horizon. (ii) The observation and state spaces are often continuous, which induces a sample complexity that scales exponentially with the extrinsic dimension. Addressing such challenges requires learning a minimal but sufficient representation of the observation and state histories by exploiting the structure of the POMDP.

To this end, we propose a reinforcement learning algorithm named Represent to Control (RTC), which learns the representation at two levels while optimizing the policy. (i) For each step, RTC learns to represent the state with a low-dimensional feature, which factorizes the transition kernel. (ii) Across multiple steps, RTC learns to represent the full history with a low-dimensional embedding, which assembles the per-step feature. We integrate (i) and (ii) in a unified framework that allows a variety of estimators (including maximum likelihood estimators and generative adversarial networks). For a class of POMDPs with a low-rank structure in the transition kernel, RTC attains an $O(1/\epsilon^2)$ sample complexity that scales polynomially with the horizon and the intrinsic dimension (that is, the rank). Here $\epsilon$ is the optimality gap. To our best knowledge, RTC is the first sample-efficient algorithm that bridges representation learning and policy optimization in POMDPs with infinite observation and state spaces.

## 1 INTRODUCTION

Deep reinforcement learning demonstrates significant empirical successes in Markov decision processes (MDPs) with large state spaces (Mnih et al., 2013; 2015; Silver et al., 2016; 2017). Such empirical successes are attributed to the integration of representation learning into reinforcement learning. In other words, mapping the state to a low-dimensional feature enables model/value learning and optimal control in a sample-efficient manner. Meanwhile, it becomes more theoretically understood that the low-dimensional feature is the key to sample efficiency in the linear setting (Cai

---

[*]Equal Contribution.

et al., 2020; Jin et al., 2020b; Ayoub et al., 2020; Agarwal et al., 2020; Modi et al., 2021; Uehara et al., 2021).

In contrast, partially observed Markov decision processes (POMDPs) with large observation and state spaces remain significantly more challenging. Due to a lack of the Markov property, the low-dimensional feature of the observation at each step is insufficient for the prediction and control of the future (Sondik, 1971; Papadimitriou and Tsitsiklis, 1987; Coates et al., 2008; Azizzadenesheli et al., 2016; Guo et al., 2016). Instead, it is necessary to obtain a low-dimensional embedding of the history, which assembles the low-dimensional features across multiple steps (Hefny et al., 2015; Sun et al., 2016). In practice, learning such features and embeddings requires various heuristics, e.g., recurrent neural network architectures and auxiliary tasks (Hausknecht and Stone, 2015; Li et al., 2015; Mirowski et al., 2016; Girin et al., 2020). In theory, the best results are restricted to the tabular setting (Azizzadenesheli et al., 2016; Guo et al., 2016; Jin et al., 2020a; Liu et al., 2022), which does not involve representation learning.

To this end, we identify a class of POMDPs with a low-rank structure on the state transition kernel (but not on the observation emission kernel), which allows prediction and control in a sample-efficient manner. More specifically, the transition admits a low-rank factorization into two unknown features, whose dimension is the rank. On top of the low-rank transition, we define a Bellman operator, which performs a forward update for any finite-length trajectory. The Bellman operator allows us to further factorize the history across multiple steps to obtain its embedding, which assembles the per-step feature.

By integrating the two levels of representation learning, that is, (i) feature learning at each step and (ii) embedding learning across multiple steps, we propose a sample-efficient algorithm, namely Represent to Control (RTC), for POMDPs with infinite observation and state spaces. The key to RTC is balancing exploitation and exploration along the representation learning process. To this end, we construct a confidence set of embeddings upon identifying and estimating the Bellman operator, which further allows efficient exploration via optimistic planning. It is worth mentioning that such a unified framework allows a variety of estimators (including maximum likelihood estimators and generative adversarial networks).

We analyze the sample efficiency of RTC under the future and past sufficiency assumptions. In particular, such assumptions ensure that the future and past observations are sufficient for identifying the belief state, which captures the information-theoretic difficulty of POMDPs. We prove that RTC attains an $O(1/\epsilon^2)$ sample complexity that scales polynomially with the horizon and the dimension of the feature (that is, the rank of the transition). Here $\epsilon$ is the optimality gap. The polynomial dependency on the horizon is attributed to embedding learning across multiple steps, while polynomial dependency on the dimension is attributed to feature learning at each step, which is the key to bypassing the infinite sizes of the observation and state spaces.

**Contributions.** In summary, our contribution is threefold.

- We identify a class of POMDPs with the low-rank transition, which allows representation learning and reinforcement learning in a sample-efficient manner.
- We propose RTC, a principled approach integrating embedding and control in the low-rank POMDP.
- We establish the sample efficiency of RTC in the low-rank POMDP with infinite observation and state spaces.

**Related Work.** Our work follows the previous studies of POMDPs. In general, solving a POMDP is intractable from both the computational and the statistical perspectives (Papadimitriou and Tsitsiklis, 1987; Vlassis et al., 2012; Azizzadenesheli et al., 2016; Guo et al., 2016; Jin et al., 2020a). Given such computational and statistical barriers, previous works attempt to identify tractable POMDPs. In particular, Azizzadenesheli et al. (2016); Guo et al. (2016); Jin et al. (2020a); Liu et al. (2022) consider the tabular POMDPs with (left) invertible emission matrices. Efroni et al. (2022) considers the POMDPs where the state is fully determined by the most recent observations of a fixed length. Cayci et al. (2022) analyze POMDPs where a finite internal state can approximately determine the state. In contrast, we analyze POMDPs with the low-rank transition and allow the state and observation spaces to be arbitrarily large. Meanwhile, our analysis hinges on the future and past sufficiency assumptions, which only require that the density of the state is identified by that of the future and

past observations, respectively. In recent work, Cai et al. (2022) also utilizes the low-rank property in the transition. Nevertheless, Cai et al. (2022) assumes that the feature representation of state-action pairs is known, thus relieving the agent from feature learning. In contrast, we aim to recover the efficient state-action representation for planning. In terms of the necessity of exploration, Aziz-zadenesheli et al. (2016); Guo et al. (2016) analyze POMDPs where an arbitrary policy can conduct efficient exploration. Similarly, Cayci et al. (2022) consider POMDPs with a finite concentrability coefficient (Munos, 2003; Chen and Jiang, 2019), where the visitation density of an arbitrary policy is close to that of the optimal policy. In contrast, Jin et al. (2020a); Efroni et al. (2022); Cai et al. (2022) consider POMDPs where strategic exploration is necessary. In our work, we follow Jin et al. (2020a); Efroni et al. (2022); Cai et al. (2022) and design strategic exploration to attain sample efficiency in solving the POMDPs. Our work is also related to the previous study of MDP with rich observations, where the authors propose to recover a possibly finite latent state of the observations (Misra et al., 2020; Zhang et al., 2022). In contrast, we propose to recover the latent state based on interaction history. In addition, our work conducts latent recovery under the more challenging POMDP setup. See also §B for additional literature review on related study of latent state space models and MDPs.

**Notation** We denote by $\mathbb{R}^d_+$ the space of $d$-dimensional vectors with nonnegative entries. We denote by $L^p(\mathcal{X})$ the $L^p$ space of functions defined on $\mathcal{X}$. We denote by $\Delta(d)$ the space of $d$-dimensional probability density arrays, namely, the $d$-dimensional nonnegative arrays that sums up to one. We denote by $[H] = \{1, \ldots, H\}$ the index set of size $H$. For a linear operator $M$ mapping from an $L^p$ space to an $L^q$ space, we denote by $\|M\|_{p \mapsto q}$ the operator norm of $M$. For a vector $x \in \mathbb{R}^d$, we denote by $[x]_i$ the $i$-th entry of $x$.

## 2 PARTIALLY OBSERVABLE MARKOV DECISION PROCESS

We define a partially observable Markov decision process (POMDP) by the following tuple,
$$\mathcal{M} = (\mathcal{S}, \mathcal{A}, \mathcal{O}, \{\mathbb{P}_h\}_{h \in [H]}, \{\mathbb{O}_h\}_{h \in [H]}, r, H, \mu_1),$$
where $H$ is the length of an episode, $\mu_1$ is the initial distribution of state $s_1$, and $\mathcal{S}$, $\mathcal{A}$, $\mathcal{O}$ are the state, action, and observation spaces, respectively. Here $\mathbb{P}_h(\cdot \,|\, \cdot, \cdot)$ is the transition kernel, $\mathbb{O}_h(\cdot \,|\, \cdot)$ is the emission kernel, and $r(\cdot)$ is the reward function. In each episode, the agent with the policy $\pi = \{\pi_h\}_{h \in [H]}$ interact with the environment as follows. The environment select an initial state $s_1$ drawn from the distribution $\mu_1$. In the $h$-th step, the agent receives the reward $r(o_h)$ and the observation $o_h$ drawn from the observation density $\mathbb{O}_h(\cdot \,|\, s_h)$, and makes the decision $a_h = \pi_h(\tau_1^h)$ according to the policy $\pi_h$, where $\tau_1^h = \{o_1, a_1, \ldots, a_{h-1}, o_h\}$ is the interaction history. The environment then transits into the next state $s_{h+1}$ drawn from the transition distribution $\mathbb{P}_h(\cdot \,|\, s_h, a_h)$. The procedure terminates until the environment transits into the termination state $s_{H+1}$.

In the sequel, we assume that the action space $\mathcal{A}$ is finite with capacity $|\mathcal{A}| = A$. Meanwhile, we highlight that the observation and state spaces $\mathcal{O}$ and $\mathcal{S}$ are possibly infinite.

**Value Functions and Learning Objective.** For a given policy $\pi = \{\pi_h\}_{h \in [H]}$, we define the following value function that captures the expected cumulative rewards from interactions,
$$V^\pi = \mathbb{E}_\pi \left[ \sum_{h=1}^H r(o_h) \right]. \tag{2.1}$$
Here we denote by $\mathbb{E}_\pi$ the expectation taken with respect to the policy $\pi$, the transition dynamics, and the emission. Our goal is to derive a policy that maximizes the cumulative rewards. In particular, we aim to derive the $\epsilon$-suboptimal policy $\pi$ such that
$$V^{\pi^*} - V^\pi \le \epsilon,$$
based on minimal interactions with the environment, where $\pi^* = \mathrm{argmax}_\pi V^\pi$ is the optimal policy.

**Notations of POMDP.** In the sequel, we introduce notations of the POMDP to simplify the discussion. We define
$$a_h^{h+k-1} = \{a_h, a_{h+1}, \ldots, a_{h+k-1}\}, \quad o_h^{h+k} = \{o_h, o_{h+1}, \ldots, o_{h+k}\}$$
as the sequences of actions and observations, respectively. Correspondingly, we write $r(o_1^H) = \sum_{h=1}^H r(o_h)$ as the cumulative rewards for the observation sequence $o_1^H$. Meanwhile, we denote by

$\tau_h^{h+k}$ the sequence of interactions from the $h$-th step to the $(h+k)$-th step, namely,

$$\tau_h^{h+k} = \{o_h, a_h, \ldots, o_{h+k-1}, a_{h+k-1}, o_{h+k}\} = \{a_h^{h+k-1}, o_h^{h+k}\}.$$

Similarly, we denote by $\underline{\tau}_h^{h+k}$ the sequence of interactions from the $h$-th step to the $(h+k)$-th step that includes the latest action $a_{h+k}$, namely,

$$\underline{\tau}_h^{h+k} = \{o_h, a_h, \ldots, o_{h+k}, a_{h+k}\} = \{a_h^{h+k}, o_h^{h+k}\}.$$

In addition, with a slight abuse of notation, we define

$$\mathbb{P}^\pi(\tau_h^{h+k}) = \mathbb{P}^\pi(o_h, \ldots, o_{h+k} \mid a_h, \ldots, a_{h+k-1}) = \mathbb{P}^\pi(o_h^{h+k} \mid a_h^{h+k-1}),$$

$$\mathbb{P}^\pi(\tau_h^{h+k} \mid s_h) = \mathbb{P}^\pi(o_h, \ldots, o_{h+k} \mid s_h, a_h, \ldots, a_{h+k-1}) = \mathbb{P}^\pi(o_h^{h+k} \mid s_h, a_h^{h+k-1}).$$

**Extended POMDP.** To simplify the discussion and notations in our work, we introduce an extension of the POMDP, which allows us to access steps $h$ smaller than zero and larger than the length $H$ of an episode.

In particular, the interaction of an agent with the extended POMDP starts with a dummy initial state $s_{1-\ell}$ for some $\ell > 0$. During the interactions, all the dummy action and observation sequences $\underline{\tau}_{1-\ell}^0 = \{o_{1-\ell}, a_{1-\ell}, \ldots, o_0, a_0\}$ leads to the same initial state distribution $\mu_1$ that defines the POMDP. Moreover, the agent is allowed to interact with the environment for $k$ steps after observing the final observation $o_H$ of an episode. Nevertheless, the agent only collects the reward $r(o_h)$ at steps $h \in [H]$, which leads to the same learning objective as the POMDP. In addition, we denote by $[H]^+ = \{1 - \ell, \ldots, H + k\}$ the set of steps in the extended POMDP. In the sequel, we do not distinguish between a POMDP and an extended POMDP for the simplicity of presentation.

# 3 A SUFFICIENT EMBEDDING FOR PREDICTION AND CONTROL

The key of solving a POMDP is the practice of inference, which recovers the density or linear functionals of density (e.g., the value functions) of future observation given the interaction history. To this end, previous approaches (Shani et al., 2013) typically maintain a belief, namely, a conditional density $\mathbb{P}(s_h = \cdot \mid \tau_1^h)$ of the current state given the interaction history. The typical inference procedure first conducts filtering, namely, calculating the belief at $(h+1)$-th step given the belief at $h$-th step. Upon collecting the belief, the density of future observation is obtained via prediction, which acquires the distribution of future observations based on the distribution of state $s_{h+1}$.

In the case that maintaining a belief or conducting the prediction is intractable, previous approaches establish a predictive state (Hefny et al., 2015; Sun et al., 2016), which is an embedding that is sufficient for inferring the density of future observations given the interaction history. Such approaches typically recover the filtering of predictive representations by solving moment equations. In particular, Hefny et al. (2015); Sun et al. (2016) establishes such moment equations based on structural assumptions on the filtering of such predictive states. Similarly, Anandkumar et al. (2012); Jin et al. (2020a) establishes a sequence of observation operators and recovers the trajectory density via such observation operators.

Motivated by the previous work, we aim to construct a embedding that are both learn-able and sufficient for control. A sufficient embedding for control is the density of the trajectory, namely,

$$\Phi(\tau_1^H) = \mathbb{P}(\tau_1^H). \tag{3.1}$$

Such an embedding is sufficient as it allows us to estimate the cumulative rewards function $V^\pi$ of an arbitrary given policy $\pi$. In the sequel, we aims to estimate such an embedding and further conduct planning based on the estimated embedding. Nevertheless, estimating such an embedding is challenging when the length $H$ of an episode and the observation space $\mathcal{O}$ are large. To this end, we exploit the low-rank structure in the state transition of POMDPs.

## 3.1 LOW-RANK POMDP

**Assumption 3.1** (Low-Rank POMDP). We assume that the transition kernel $\mathbb{P}_h$ takes the following low-rank form for all $h \in [H]^+$,

$$\mathbb{P}_h(s_{h+1} \mid s_h, a_h) = \psi_h^*(s_{h+1})^\top \phi_h^*(s_h, a_h),$$

where

$$\psi_h^* : \mathcal{S} \mapsto \mathbb{R}_+^d, \quad \phi_h^* : \mathcal{S} \times \mathcal{A} \mapsto \Delta(d)$$

are unknown features.

Here recall that we denote by $[H]^+ = \{1 - \ell, \ldots, H + k\}$ the set of steps in the extended POMDP. Note that our low-rank POMDP assumption does not specify the form of emission kernels. In contrast, we only require the transition kernels of states to be linear in unknown features.

**Function Approximation.** We highlight that the features in Assumption 3.1 are unknown to us. Correspondingly, we assume that we have access to a parameter space $\Theta$ that allows us to fit such features as follows.

**Definition 3.2** (Function Approximation). We define the following function approximation space $\mathcal{F}^\Theta = \{\mathcal{F}_h^\Theta\}_{h \in [H]}$ corresponding to the parameter space $\Theta$,

$$\mathcal{F}_h^\Theta = \big\{(\psi_h^\theta, \phi_h^\theta, \mathbb{O}_h^\theta) : \theta \in \Theta\big\}, \quad \forall h \in [H]^+.$$

Here, $\mathbb{O}_h^\theta : \mathcal{S} \times \mathcal{O} \mapsto \mathbb{R}_+$ is an emission kernel and $\psi_h^\theta : \mathcal{S} \mapsto \mathbb{R}_+^d$, $\phi_h^\theta : \mathcal{S} \mapsto \Delta(d)$ are features for all $h \in [H]^+$ and $\theta \in \Theta$. In addition, it holds that $\psi^\theta(\cdot)^\top \phi^\theta(s_h, a_h)$ defines a probability over $s_{h+1} \in \mathcal{S}$ for all $h \in [H]^+$ and $(s_h, a_h) \in \mathcal{S} \times \mathcal{A}$.

Here we denote by $\psi_h^\theta, \phi_h^\theta, \mathbb{O}_h^\theta$ a parameterization of features and emission kernels. In practice, one typically utilizes linear or neural network parameterization for the features and emission kernels. In the sequel, we write $\mathbb{P}^\theta$ and $\mathbb{P}^{\theta,\pi}$ as the probability densities corresponding to the transition dynamics defined by $\{\psi_h^\theta, \phi_h^\theta, \mathbb{O}_h^\theta\}_{h \in [H]}$ and policy $\pi$, respectively. We impose the following realizability assumption to ensure that the true model belongs to the parameterized function space $\mathcal{F}^\Theta$.

**Assumption 3.3** (Realizable Parameterization). We assume that there exists a parameter $\theta^* \in \Theta$, such that $\psi_h^{\theta^*} = \psi_h^*$, $\phi_h^{\theta^*} = \phi_h^*$, and $\mathbb{O}_h^{\theta^*} = \mathbb{O}_h$ for all $h \in [H]$.

We define the following forward emission operator as a generalization of the emission kernel.

**Definition 3.4** (Forward Emission Operator). We define the following forward emission operator $\mathbb{U}_h^\theta : L^1(\mathcal{S}) \mapsto L^1(\mathcal{A}^k \times \mathcal{O}^{k+1})$ for all $h \in [H]$,

$$(\mathbb{U}_h^\theta f)(\tau_h^{h+k}) = \int_{\mathcal{S}} \mathbb{P}^\theta(\tau_h^{h+k} \,|\, s_h) \cdot f(s_h) \mathrm{d}s_h, \quad \forall f \in L^1(\mathcal{S}), \, \forall \tau_h^{h+k} \in \mathcal{A}^k \times \mathcal{O}^{k+1}. \qquad (3.2)$$

Here recall that we denote by $\tau_h^{h+k} = \{a_h^{h+k-1}, o_h^{h+k}\} \in \mathcal{A}^k \times \mathcal{O}^{k+1}$ the trajectory of interactions. In addition, recall that we define $\mathbb{P}^\theta(\tau_h^k \,|\, s_h) = \mathbb{P}^\theta(o_h^{h+k} \,|\, s_h, a_h^{h+k-1})$ for notational simplicity. We remark that here we omit the dependency of $\mathbb{U}_h^\theta$ on the length $k$ of trajectory to simplify the notation.

We remark that when applying to a belief or a density over state $s_h$, the forward emission operator returns the density of trajectory $\tau_h^{h+k}$ of $k$ steps ahead of the $h$-th step.

**Bottleneck Factor Interpretation of Low-Rank Transition.** Recall that in Assumption 3.1, the feature $\phi_h^*$ maps from the state-action pair $(s_h, a_h) \in \mathcal{S} \times \mathcal{A}$ to a $d$-dimensional simplex in $\Delta(d)$. Equivalently, one can consider the low-rank transition as a latent variable model, where the next state $s_{h+1}$ is generated by first generating a bottleneck factor $q_h \sim \phi^*(s_h, a_h)$ and then generating the next state $s_{h+1}$ by $[\psi^*(\cdot)]_{q_h}$. In other words, the probability array $\phi^*(s_h, a_h) \in \Delta(d)$ induces a transition dynamics from the state-action pair $(s_h, a_h)$ to the bottleneck factor $q_h \in [d]$ as follows,

$$\mathbb{P}_h(q_h \,|\, s_h, a_h) = \big[\phi_h^*(s_h, a_h)\big]_{q_h}, \quad \forall q_h \in [d].$$

Correspondingly, we write $\mathbb{P}_h(s_{h+1} \,|\, q_h) = [\psi_h^*(s_{h+1})]_{q_h}$ the transition probability from the bottleneck factor $q_h \in [d]$ to the state $s_{h+1} \in \mathcal{S}$. See Figure 1 for an illustration of the data generating process with the bottleneck factors.

**Understanding Bottleneck Factor.** Utilizing the low-rank structure of the state transition requires us to understand the bottleneck factors $\{q_h\}_{h \in [H]}$ defined by the low-rank transition. We highlight that the bottleneck factor $q_h$ is a compressed and sufficient factor for inference. In particular, the bottleneck factor $q_h$ determines the distribution of next state $s_{h+1}$ through the feature $\psi_h^*(s_{h+1} = \cdot) = \mathbb{P}(s_{h+1} = \cdot \,|\, q_h = \cdot)$. Such a property motivate us to obtain our desired embedding via decomposing the density of trajectory based on the feature set $\{\psi_h^*\}_{h \in [H]^+}$. To achieve such a decomposition, we first introduce the following sufficiency condition for all the parameterized features $\psi_h^\theta$ with $\theta \in \Theta$.

**Assumption 3.5** (Future Sufficiency). We define the mapping $g_h^\theta : \mathcal{A}^k \times \mathcal{O}^{k+1} \mapsto \mathbb{R}^d$ for all parameter $\theta \in \Theta$ and $h \in [H]$ as follows,

$$g_h^\theta = \left[\mathbb{U}_h^\theta[\psi_{h-1}^\theta]_1, \ldots, \mathbb{U}_h^\theta[\psi_{h-1}^\theta]_d\right]^\top,$$

where we denote by $[\psi_{h-1}^\theta]_i$ the $i$-th entry of the mapping $\psi_{h-1}^\theta$ for all $i \in [d]$. We assume for some $k > 0$ that the matrix

$$M_h^\theta = \int_{\mathcal{A}^k \times \mathcal{O}^{k+1}} g_h^\theta(\tau_h^{h+k}) g_h^\theta(\tau_h^{h+k})^\top \mathrm{d}\tau_h^{h+k} \in \mathbb{R}^{d \times d}$$

is invertible. We denote by $M_h^{\theta,\dagger}$ the inverse of $M_h^\theta$ for all parameter $\theta \in \Theta$ and $h \in [H]$.

Intuitively, the future sufficiency condition in Assumption 3.5 guarantees that the density of trajectory $\tau_h^{h+k}$ in the future captures the information of the bottleneck variable $q_{h-1}$, which further captures the belief at the $h$-th step. To see such a fact, we have the following lemma.

**Lemma 3.6** (Pseudo-Inverse of Forward Emission). We define linear operator $\mathbb{U}_h^{\theta,\dagger} : L^1(\mathcal{A}^k \times \mathcal{O}^{k+1}) \mapsto L^1(\mathcal{S})$ for all $\theta \in \Theta$ and $h \in [H]$ as follows,

$$(\mathbb{U}_h^{\theta,\dagger} f)(s_h) = \int_{\mathcal{A}^k \times \mathcal{O}^{k+1}} \psi_{h-1}^\theta(s_h)^\top M_h^{\theta,\dagger} g_h^\theta(\tau_h^{h+k})$$
$$\cdot f(\tau_h^{h+k})\mathrm{d}\tau_h^{h+k}, \quad (3.3)$$

where $f \in L^1(\mathcal{A}^k \times \mathcal{O}^{k+1})$ is the input of linear operator $\mathbb{U}_h^{\theta,\dagger}$ and $g_h^\theta$ is the mapping defined in Assumption 3.5. Under Assumptions 3.1 and 3.5, it holds for all $h \in [H]$, $\theta \in \Theta$, and $\pi \in \Pi$ that

$$\mathbb{U}_h^{\theta,\dagger} \mathbb{U}_h^\theta(\mathbb{P}_h^{\theta,\pi}) = \mathbb{P}_h^{\theta,\pi}.$$

Here $\mathbb{P}_h^{\theta,\pi} \in L^1(\mathcal{S})$ maps from all state $s_h \in \mathcal{S}$ to the probability $\mathbb{P}_h^{\theta,\pi}(s_h)$, which is the probability of visiting the state $s_h$ in the $h$-th step when following the policy $\pi$ and the model defined by parameter $\theta$.

*Proof.* See §D.1 for a detailed proof. □

Figure 1: Directed acyclic graph (DAG) of a POMDP with low-rank transition. Here $\{s_h, s_{h+1}\}$, $\{o_h, o_{h+1}\}$, $a_h$, $r_h$ are the states, observations, action, and reward, respectively. In addition, we denote by $q_h$ the bottleneck factor induced by the low-rank transition, which depends on the state and action pair $(s_h, a_h)$ and determines the density of next state $s_{h+1}$. In the DAG, we represent observable and unobservable variables by the shaded and unshaded nodes, respectively. In addition, we use the dashed node and arrows for the latent factor $q_h$ and its corresponding transitions, respectively.

By Lemma 3.6, the forward emission operator $\mathbb{U}_h^\theta$ defined in Definition 3.4 has a pseudo-inverse $\mathbb{U}_h^{\theta,\dagger}$ under the future sufficiency condition in Assumption 3.5. Thus, one can identify the belief state by inverting the conditional density of the trajectory $\tau_h^{h+k}$ given the interaction history $\tau_1^h$. More importantly, such invertibility further allows us to decompose the desired embedding $\Phi(\tau_1^H)$ in (3.1) across steps, which we introduce in the sequel.

### 3.2 MULTI-STEP EMBEDDING DECOMPOSITION VIA BELLMAN OPERATOR

To accomplish the multi-step decomposition of embedding, we first define the Bellman operator as follows.

**Definition 3.7** (Bellman Operator). We define the Bellman operators $\mathbb{B}_h^\theta(a_h, o_h) : L^1(\mathcal{A}^k \times \mathcal{O}^{k+1}) \mapsto L^1(\mathcal{A}^k \times \mathcal{O}^{k+1})$ for all $(a_h, o_h) \in \mathcal{A} \times \mathcal{O}$ and $h \in [H]$ as follows,

$$\left(\mathbb{B}_h^\theta(a_h, o_h)f\right)(\tau_{h+1}^{h+k+1}) = \int_{\mathcal{S}} \mathbb{P}^\theta(\tau_h^{h+k+1} \,|\, s_h) \cdot (\mathbb{U}_h^{\theta,\dagger} f)(s_h)\mathrm{d}s_h, \quad \forall \tau_{h+1}^{h+k+1} \in \mathcal{A}^k \times \mathcal{O}^{k+1}.$$

Here recall that we denote by $\tau_h^{h+k+1} = \{o_h^{h+k+1}, a_h^{h+k}\}$ and $\mathbb{P}^\theta(\tau_h^{h+k+1} \,|\, s_h) = \mathbb{P}^\theta(o_h^{h+k+1} \,|\, s_h, a_h^{h+k+1})$ for notational simplicity.

We call $\mathbb{B}_h^\theta(a_h, o_h)$ in Definition 3.7 a Bellman operator as it performs a temporal transition from the density of trajectory $\tau_h^{h+k}$ to the density of trajectory $\tau_{h+1}^{h+k+1}$ and the observation $o_h$, given that

one take action $a_h$ at the $h$-th step. More specifically, Assumption 3.5 guarantees that the density of trajectory $\tau_h^{h+k}$ identifies the density of $s_h$ in the $h$-th step. The Bellman operator then performs the transition from the density of $s_h$ to the density of the trajectory $\tau_{h+1}^{h+k+1}$ and observation $o_h$ given the action $a_h$. The following Lemma shows that our desired embedding $\Phi(\tau_1^H)$ can be decomposed into products of the Bellman operators defined in Definition 3.7.

**Lemma 3.8** (Embedding Decomposition). Under Assumptions 3.1 and 3.5, it holds for all the parameter $\theta \in \Theta$ that

$$\mathbb{P}^\theta(\tau_1^H) = \frac{1}{A^k} \cdot \int_{\mathcal{A}^k \times \mathcal{O}^{k+1}} \left[ \mathbb{B}_H^\theta(o_H, a_H) \ldots \mathbb{B}_1^\theta(o_1, a_1) b_1^\theta \right] (\tau_{H+1}^{H+k+1}) \mathrm{d}\tau_{H+1}^{H+k+1}.$$

Here recall that we denote by $\tau_{H+1}^{H+k+1} = \{a_{H+1}^{H+k}, o_{H+1}^{H+k+1}\}$ the dummy future trajectory. Meanwhile, we define the following initial trajectory density,

$$b_1^\theta(\tau_1^k) = \mathbb{U}_1^\theta \mu_1 = \mathbb{P}^\theta(\tau_1^k), \quad \forall \tau_1^k \in \mathcal{A}^k \times \mathcal{O}^{k+1}.$$

*Proof.* See §D.3 for a detailed proof. □

By Lemma 3.8, we can obtain the desired representation $\Phi(\tau_1^H) = \mathbb{P}(\tau_1^H)$ based on the product of the Bellman operators. It now remains to estimate the Bellman operators across each step. In the sequel, we introduce an identity that allows us to recover the Bellman operators based on observations.

**Estimating Bellman Operator.** In the sequel, we introduce the following notation to simplify our discussion,

$$z_h = \tau_h^{h+k} = \{o_h, a_h, \ldots, a_{h+k-1}, o_{h+k}\} \in \mathcal{A}^k \times \mathcal{O}^{k+1}, \tag{3.4}$$

$$w_{h-1} = \underline{\tau}_{h-\ell}^{h-1} = \{o_{h-\ell}, a_{h-\ell}, \ldots, o_{h-1}, a_{h-1}\} \in \mathcal{A}^\ell \times \mathcal{O}^\ell. \tag{3.5}$$

We first define two density mappings that induce the identity of the Bellman Operator. We define the density mapping $\mathbb{X}_h^{\theta,\pi} : \mathcal{A}^\ell \times \mathcal{O}^\ell \mapsto L^1(\mathcal{A}^k \times \mathcal{O}^{k+1})$ as follows,

$$\mathbb{X}_h^{\theta,\pi}(w_{h-1}) = \mathbb{P}^{\theta,\pi}(w_{h-1}, z_h = \cdot), \quad \forall w_{h-1} \in \mathcal{A}^\ell \times \mathcal{O}^\ell. \tag{3.6}$$

Intuitively, the density mapping $\mathbb{X}_h^{\theta,\pi}$ maps from an input trajectory $w_{h-1}$ to the density of $z_h$, which represents the density of $k$-steps interactions following the input trajectory $w_{h-1}$. Similarly, we define the density mapping $\mathbb{Y}_h^{\theta,\pi} : \mathcal{A}^{\ell+1} \times \mathcal{O}^{\ell+1} \mapsto L^1(\mathcal{A}^k \times \mathcal{O}^{k+1})$ as follows,

$$\mathbb{Y}_h^{\theta,\pi}(w_{h-1}, a_h, o_h) = \mathbb{P}^{\theta,\pi}(w_{h-1}, a_h, o_h, z_{h+1} = \cdot), \quad \forall (w_{h-1}, a_h, o_h) \in \mathcal{A}^{\ell+1} \times \mathcal{O}^{\ell+1} \tag{3.7}$$

Based on the two density mappings defined in (3.6) and (3.7), respectively, we have the following identity for all $h \in [H]$ and $\theta \in \Theta$,

$$\mathbb{B}_h^\theta(a_h, o_h) \mathbb{X}_h^{\theta,\pi}(w_{h-1}) = \mathbb{Y}_h^{\theta,\pi}(w_{h-1}, a_h, o_h), \quad \forall w_{h-1} \in \mathcal{A}^{\ell+1} \times \mathcal{O}^{\ell+1}. \tag{3.8}$$

See §D.2 for the proof of (3.8). We highlight that the identity in (3.8) allows us to estimate the Bellman operator $\mathbb{B}_h^{\theta^*}(a_h, o_h)$ under the true parameter $\theta^* \in \Theta$. In particular, both $\mathbb{X}_h^{\theta^*,\pi}$ and $\mathbb{Y}_h^{\theta^*,\pi}$ are density mappings involving the observations and actions, and can be estimated based on observable variables from the POMDP. Upon fitting such density mappings, we can recover the Bellman operator $\mathbb{B}_h^{\theta^*}(a_h, o_h)$ by solving the identity in (3.8).

**An Overview of Embedding Learning.** We now summarize the learning procedure of the embedding. First, we estimate the density mappings defined in (3.6) and (3.7) under the true parameter $\theta^*$ based on interaction history. Second, we estimate the Bellman operators $\{\mathbb{B}_h^{\theta^*}(a_h, o_h)\}_{h \in [H]}$ based on the identity in (3.8) and the estimated density mappings in the first step. Finally, we recover the embedding $\Phi(\tau_1^H)$ by assembling the Bellman operators according to Lemma 3.8.

## 4 ALGORITHM

In what follows, we present Represent to Control (RTC), an online learning algorithm that iteratively learns the embedding and conduct control based on the embedding learned. In particular, RTC iteratively fits the density mappings defined in (3.6) and (3.7) with respect to the sampling policy, and fit the Bellman operators by the identity in (3.8). Finally, RTC conducts optimistic planning by the confidence set identified in embedding learning. See §C for the detailed procedure and Algorithm 1 for a summarization of RTC.

### 4.1 DENSITY ESTIMATION

In the embedding learning, we need and estimator to recover the density mappings defined in (3.6) and (3.7). In practice, various approaches are available in fitting the density by observations. In what follows, we unify such density estimation approaches by a density estimation oracle.

**Assumption 4.1** (Density Estimation Oracle). We assume that we have access to a density estimation oracle $\mathfrak{E}(\cdot)$. Moreover, for all $\delta > 0$ and dataset $\mathcal{D}$ drawn from the density $p$ of size $n$ following a martingale process, we assume that

$$\|\mathfrak{E}(\mathcal{D}) - p\|_1 \leq C \cdot \sqrt{w_{\mathfrak{E}} \cdot \log(1/\delta)/n}$$

with probability at least $1 - \delta$. Here $C > 0$ is an absolute constant and $w_{\mathfrak{E}}$ is a parameter that depends on the density estimation oracle $\mathfrak{E}(\cdot)$.

We highlight that such convergence property can be achieved by various density estimations. In particular, when the function approximation space $\mathcal{P}$ of $\mathfrak{E}(\cdot)$ is finite, Assumption 4.1 holds for the maximum likelihood estimation (MLE) and the generative adversarial approach with $w_{\mathfrak{E}} = \log|\mathcal{P}|$ (Geer et al., 2000; Zhang, 2006; Agarwal et al., 2020). Meanwhile, $w_{\mathfrak{E}}$ scales with the entropy integral of $\mathcal{P}$ endowed with the Hellinger distance if $\mathcal{P}$ is infinite (Geer et al., 2000; Zhang, 2006). In addition, Assumption 4.1 holds for the reproducing kernel Hilbert space (RKHS) density estimation (Gretton et al., 2005; Smola et al., 2007; Cai et al., 2022) with $w_{\mathfrak{E}} = \mathrm{poly}(d)$, where $d$ is rank of the low-rank transition (Cai et al., 2022).

Upon fitting the density mappings $\widehat{\mathbb{X}}_h^t$ and $\widehat{\mathbb{Y}}_h^t$ in the $t$-th iterate, we estimate the Bellman operators by minimizing the following objective,

$$L_h^t(\theta) = \sup_{a_{h-\ell}^h \in \mathcal{A}^{\ell+1}} \int_{\mathcal{O}^{\ell+1}} \|\mathbb{B}_h^\theta(a_h, o_h)\widehat{\mathbb{X}}_h^t(w_{h-1}) - \widehat{\mathbb{Y}}_h^t(w_{h-1}, a_h, o_h)\|_1 \mathrm{d}o_{h-\ell}^h. \tag{4.1}$$

Here recall that we define the shorthand $w_{h-1} = \{o_{h-\ell}, a_{h-\ell}, \dots, o_{h-1}, a_{h-1}\}$ in (3.5).

### 4.2 OPTIMISTIC PLANNING

The learning of Bellman operators allows us to identify a confidence interval for the parameter and the associated embedding. In particular, we define the following confidence set,

$$\mathcal{C}^t = \left\{\theta \in \Theta : \max\{\|b_1^\theta - \widehat{b}_1^t\|_1, L_h^t(\theta)\} \leq \beta_t \cdot \sqrt{1/t}, \quad \forall h \in [H]\right\}, \tag{4.2}$$

where $\beta_t$ is the tuning parameter in the $t$-th iterate. To conduct optimistic planning, we seek for the policy that maximizes the return among all parameters $\theta \in \mathcal{C}^t$ and the corresponding features. The update of policy takes the following form,

$$\pi^t \leftarrow \underset{\pi \in \Pi}{\mathrm{argmax}} \max_{\theta \in \mathcal{C}^t} V^\pi(\theta).$$

Here $V^\pi(\theta)$ is the cumulative rewards estimated based on the embedding induced by $\theta$. See §C for the details.

## 5 ANALYSIS

In what follows, we present the sample complexity analysis of RTC presented in Algorithm 1. Our analysis hinges on the following assumptions.

**Assumption 5.1** (Bounded Pseudo-Inverse). We assume that $\|\mathbb{U}_h^{\theta,\dagger}\|_{1 \mapsto 1} \leq \nu$ for all $\theta \in \Theta$ and $h \in [H]$, where $\nu > 0$ is an absolute constant.

We remark that the upper bound of the pseudo-inverse in Assumption 5.1 quantifies the fundamental difficulty of solving the POMDP. In particular, the pseudo-inverse of forward emission recovers the state density at the $h$-th step based on the trajectory $\tau_h^{h+k}$ from the $h$-th step to the $(h+k)$-th step. Thus, the upper bound $\nu$ on such pseudo-inverse operator characterizes how ill-conditioned the belief recovery task is based on the trajectory $\tau_h^{h+k}$. In what follows, we impose a similar past sufficiency assumption.

---

**Algorithm 1** Represent to Control

---

**Require:** Number of iterates $T$. A set of tuning parameters $\{\beta_t\}_{t \in [T]}$.

1: **Initialization:** Set $\pi_0$ as a deterministic policy. Set the dataset $\mathcal{D}_h^0(a_{h-\ell}^{h+k})$ as an empty set for all $(h, a_{h-\ell}^{h+k}) \in [H] \times \mathcal{A}^{k+\ell+1}$.

2: **for** $t \in [T]$ **do**

3:    **for** $(h, a_{h-\ell}^{h+k}) \in [H] \times \mathcal{A}^{k+\ell+1}$ **do**

4:       Start a new episode from the $(1 - \ell)$-th step.

5:       Execute policy $\pi^{t-1}$ until the $(h - \ell)$-th step and receive the observations $o_{1-\ell}^{h-\ell}$.

6:       Execute the action sequence $a_{h-\ell}^{h+k}$ regardless of the observations and receive the observations $o_{h-\ell+1}^{h+k+1}$.

7:       Update the dataset $\mathcal{D}_h^t(a_{h-\ell}^{h+k}) \leftarrow \mathcal{D}_h^{t-1}(a_{h-\ell}^{h+k}) \cup \{o_{h-\ell}^{h+k+1}\}$.

8:    **end for**

9:    Estimate the density of trajectory $\widehat{\mathbb{P}}_h^t(\cdot \,|\, a_{h-\ell}^{h+k}) \leftarrow \mathfrak{E}\big(\mathcal{D}^t(a_{h-\ell}^{h+k})\big)$ for all $h \in [H]$.

10:    Update the density mappings $\widehat{\mathbb{X}}_h^t$ and $\widehat{\mathbb{Y}}_h^t$ as follows,
$$\widehat{\mathbb{X}}_h^t(w_{h-1}) = \widehat{\mathbb{P}}_h^t(w_{h-1}, z_h = \cdot), \qquad \widehat{\mathbb{Y}}_h^t(w_{h-1}, a_h, o_h) = \widehat{\mathbb{P}}_h^t(w_{h-1}, a_h, o_h, z_{h+1} = \cdot).$$

11:    Update the initial density estimation $\widehat{b}_1^t(\tau_1^H) \leftarrow \widehat{\mathbb{P}}^t(\tau_1^H)$.

12:    Update the confidence set $\mathcal{C}^t$ by (4.2).

13:    Update the policy $\pi^t \leftarrow \operatorname{argmax}_{\pi \in \Pi} \max_{\theta \in \mathcal{C}^t} V^\pi(\theta)$.

14: **end for**

15: **Output:** policy set $\{\pi^t\}_{t \in [T]}$.

---

**Assumption 5.2** (Past Sufficiency). We define for all $h \in [H]$ the following reverse emission operator $\mathbb{F}_h^{\theta,\pi} : \mathbb{R}^d \mapsto L^1(\mathcal{O}^\ell \times \mathcal{A}^\ell)$ for all $h \in [H]$, $\pi \in \Pi$, and $\theta \in \Theta$,

$$(\mathbb{F}_h^{\theta,\pi} v)(\underline{\tau}_{h-\ell}^{h-1}) = \sum_{q_{h-1} \in [d]} [v]_{q_{h-1}} \cdot \mathbb{P}^{\theta,\pi}(o_{h-\ell}^{h-1} \,|\, q_{h-1}, a_{h-\ell}^{h-1}), \quad \forall v \in \mathbb{R}^d,$$

where $(\underline{\tau}_{h-\ell}^{h-1}) \in \mathcal{A}^\ell \times \mathcal{O}^\ell$. We assume for some $\ell > 0$ that the operator $\mathbb{F}_h^{\theta,\pi}$ is left invertible for all $h \in [H]$, $\pi \in \Pi$, and $\theta \in \Theta$. We denote by $\mathbb{F}_h^{\theta,\pi,\dagger}$ the left inverse of $\mathbb{F}_h^{\theta,\pi}$. We assume further that $\|\mathbb{F}_h^{\theta,\pi,\dagger}\|_{1 \mapsto 1} \le \gamma$ for all $h \in [H]$, $\pi \in \Pi$, and $\theta \in \Theta$, where $\gamma > 0$ is an absolute constant.

We remark that the left inverse $\mathbb{F}_h^{\theta,\pi,\dagger}$ of reverse emission operator $\mathbb{F}_h^{\theta,\pi}$ recovers the density of the bottleneck factor $q_{h-1}$ based on the density of trajectory $\underline{\tau}_{h-\ell}^{h-1}$ from the $(h-\ell)$-th step to the $(h-1)$-th step. Intuitively, the past sufficiency assumption in Assumption 5.2 guarantees that the density of trajectory $\underline{\tau}_{h-\ell}^{h-1}$ from the past captures sufficient information of the bottleneck factor $q_{h-1}$, which further determines the state distribution at the $h$-th step. Thus, similar to the upper bound $\nu$ in Assumption 5.1, the upper bound $\gamma$ in Assumption 5.2 characterizes how ill-conditioned the belief recovery task is based on the trajectory $\underline{\tau}_{h-\ell}^{h-1}$ generated by the policy $\pi$.

In what follows, we analyze the mixture policy $\overline{\pi}^T$ of the policy set $\{\pi^t\}_{t \in [T]}$ returned by RTC in Algorithmn 1. In particular, the mixture policy $\overline{\pi}^T$ is executed by first sampling a policy $\pi$ uniformly from the policy set $\{\pi^t\}_{t \in [T]}$ in the beginning of an episode, and then executing $\pi$ throughout the episode.

**Theorem 5.3.** Let $\overline{\pi}^T$ be the mixture policy of the policy set $\{\pi^t\}_{t \in [T]}$ returned by Algorithm 1. Let $\beta_t = (\nu + 1) \cdot A^{2k} \cdot \sqrt{w_{\mathfrak{E}} \cdot (k + \ell) \cdot \log(H \cdot A \cdot T)}$ for all $t \in [T]$ and

$$T = \mathcal{O}\big(\gamma^2 \cdot \nu^4 \cdot d^2 \cdot w_{\mathfrak{E}}^2 \cdot H^2 \cdot A^{2(2k+\ell)} \cdot (k + \ell) \cdot \log(H \cdot A/\epsilon)/\epsilon^2\big).$$

Under Assumptions 3.1, 3.5, 4.1, 5.1, and 5.2, it holds with probability at least $1 - \delta$ that $\overline{\pi}^T$ is $\epsilon$-suboptimal.

*Proof.* See §E.3 for a detailed proof. $\qquad\square$

In Theorem 5.3, we fix the lengths of future and past trajectories $k$ and $\ell$, respectively, such that Assumptions 3.5 and 5.2 holds. Theorem 5.3 shows that the mixture policy $\overline{\pi}^T$ of the policy set

$\{\pi^t\}_{t\in[T]}$ returned by RTC is $\epsilon$-suboptimal if the number of iterations $T$ scales with $\mathcal{O}(1/\epsilon^2)$. We remark that such a dependency regarding $\epsilon$ is information-therotically optimal for reinforcement learning in MDPs (Ayoub et al., 2020; Agarwal et al., 2020; Modi et al., 2021; Uehara et al., 2021), which is a special case of POMDPs. In addition, the sample complexity $T$ depends polynomially on the length of horizon $H$, number of actions $A$, the dimension $d$ of the low-rank transition in Assumption 3.1, and the upper bounds $\nu$ and $\gamma$ in Assumptions 5.1 and 5.2, respectively. We highlight that the sample complexity depends on the observation and state spaces only through the dimension $d$ of the low-rank transition, extending the previous sample efficiency analysis of tabular POMDPs (Azizzadenesheli et al., 2016; Jin et al., 2020a). In addition, the sample complexity depends on the upper bounds of the operator norms $\nu$ and $\gamma$ in Assumptions 5.1 and 5.2, respectively, which quantify the fundamental difficulty of solving the POMDP. See §G for the analysis under the tabular POMDP setting.

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
