# OpenReview forum: "Represent to Control Partially Observed Systems: Representation Learning with Provable Sample Efficiency"
_ICLR.cc/2023/Conference — ICLR 2023 poster_

### Official Review · Reviewer_nfDQ · 2022-10-22

**Confidence:** 3
**Correctness:** 3
**Technical Novelty And Significance:** 3
**Empirical Novelty And Significance:** Not applicable
**Recommendation:** 6

**Clarity, Quality, Novelty And Reproducibility:**

The paper is not written very clearly and is not easy to follow. The use of terminology does not appear to be standard. The transition probability function of the POMDP is called kernel for unknown reasons, and the two functions it is factored into are called features. Since both terms mean something else in the ML literature, their use in this sense is quite confusing.

The choice of name for the algorithm, embed to control (ETC), is also confusing, because a well known paper on a very similar topic from 7 years ago literally has this name in its title:

Watter, M., Springenberg, J., Boedecker, J., & Riedmiller, M. (2015). Embed to control: A locally linear latent dynamics model for control from raw images. Advances in neural information processing systems, 28.

And, there is nothing to reproduce in this paper, as there are no computational experiments in it.

**Strength And Weaknesses:**

The paper addresses a central problem in learning suitable models of partially observable environments, particularly when the observation space is large and there is reason to believe that the actual system's state is relatively low dimensional. The presented approach revolves around learning low-level representations of the state, and is thus very suitable for this conference. The idea to factor the transition function into a product using a bottleneck layer (a low-dimensional probability simplex) is appealing, and reminiscent of subspace identification algorithms in the area of linear system identification that use SVD to factor observed data.

However, there is no empirical evaluation of the advantages of the proposed approach, so we cannot be sure how effective or even feasible this idea is. Absent such a verification, readers would have very little incentive to understand the mathematical analysis and implement the idea in practice. Because of this, I think the paper is still unfinished.

Some minor typos:

P.4 "a embedding" -> "an embedding"
P.4 "learn-able" -> "learnable"
P.4: "we aims" -> "we aim"
P.5: "a property motivate us" -> "a property motivates us"


**Summary Of The Paper:**

The paper presents an approach for learning POMDP models from data based on the idea of factoring the transition function of the POMDP into a product of two mappings, going trough a low-dimensional bottleneck layer that will effectively represent the hidden state. Although this is a sound idea, there is no empirical verification of how it might work in practice, in comparison with other existing methods for learning POMDPs.

**Summary Of The Review:**

Although based on an appealing idea, it is hard to say if this method would be useful in practice. Some kind of empirical evaluation would help a lot to persuade readers to adopt this method.

---

> ### Author Response · Authors · 2022-11-17
> **Author response**
>
> Thank you very much for your valuable comments. We address the concerns as follows.
>
> >The choice of name for the algorithm, embed to control (ETC), is also confusing, because a well known paper on a very similar topic from 7 years ago literally has this name in its title.
>
> Regarding the name of our algorithm:
>
> We will change the name of our paper in our revision. Our plan is to change the name to “represent to control”. Please let us know if the name is appropriate for you.
>
> >The use of terminology does not appear to be standard. The transition probability function of the POMDP is called kernel for unknown reasons, and the two functions it is factored into are called features.
>
> Regarding the terminology:
>
> Utilizing the term “transition kernel” is in fact a common practice in the RL literature and the study of stochastic processes (e.g., [1]). Meanwhile, the meaning of “feature” in ML typically changes in accordance with the context that it appears. In our cases, the “features” extract the essential information from the state action pairs in the inference of next-step prediction. We find our utilization of “features” consistent with the majority of previous RL literature (e.g. [2]).
>
> >And, there is nothing to reproduce in this paper, as there are no computational experiments in it.
>
> Regarding the Reproducibility:
>
> We remark that we justified the performance of our proposed approach from the theoretical side, which is more convincing than experiments on particular environments. Meanwhile, we highlight that our objective is exploring solvable POMDPs with optimal sample complexity, which aligns with a line of important research in POMDPs (e.g., [3][4][5]). Achieving SOTA in popular simulation environments is not the focus of our work, and does not contribute to our objective.
>
> [1] Klenke, Achim (2008). Probability Theory.
>
> [2] Jin, Chi, et al. "Provably efficient reinforcement learning with linear function approximation." Conference on Learning Theory. PMLR, 2020.
>
> [3] Liu, Qinghua, et al. "When Is Partially Observable Reinforcement Learning Not Scary?." arXiv preprint arXiv:2204.08967 (2022).
>
> [4] Jin, Chi, et al. "Sample-efficient reinforcement learning of undercomplete POMDPs." Advances in Neural Information Processing Systems 33 (2020): 18530-18539.
>
> [5] Efroni, Yonathan, et al. "Provable reinforcement learning with a short-term memory." arXiv preprint arXiv:2202.03983 (2022).

---

> > ### Comment · Reviewer_nfDQ · 2022-11-17
> > **Response to authors' response**
> >
> > The name "represent to control" looks appropriate, and using it would avoid confusion with prior work. I still find the absence of an empirical evaluation to be a big shortcoming, and grounds for rejecting the paper. Other reviewers have pointed out that several strong assumptions have been made, and potentially unfavorable (exponential) scalability wrt window sizes. These might or might not be detrimental to the applicability of the algorithm in practice, and the practical and widely accepted way to verify this is to provide an empirical study. Again, missing that, I don't think many readers would be interested in adopting the algorithm.

---

> > > ### Author Response · Authors · 2022-11-18
> > > **Author response**
> > >
> > > Thank you very much for your valuable comments. We address the concerns as follows.
> > >
> > > > I still find the absence of an empirical evaluation to be a big shortcoming, and grounds for rejecting the paper.
> > >
> > > Regarding Numerical Verification
> > >
> > > The theoretical understanding of sample efficient exploration in POMDP is at an early stage and all existing papers do not have any numerical verification yet (e.g., [1] and [2].). We agree that it is necessary to bridge theory with practice eventually but we do not have any principled algorithm on the theory side at all, which is our primary focus. In particular, we focus on the continuous setting with unknown feature representations, whose provable statistical and computational guarantees remain open before our current paper. We only address the statistical aspect as a first step, while the computational aspect is still challenging due to several hardness results even in the discrete setting (e.g., [1].). We don't want to argue the practicality of our algorithm. Instead, our algorithm is in some sense information theoretical -- our work identifies a statistically tractable subclass of POMDPs and the algorithm serves as a certificate for the regret upper bound. Identifying tractable POMDP instances is pivotal as such a class of problems is information-theoretic intractable in the worst case (e.g., [1])
> > >
> > > There is a long way toward bridging a principled algorithm with practice. One such example of MDP (a special case of POMDP) is the UCRL family of algorithms, which handles the discrete setting and is extended to the continuous setting several years later. One may argue about the practicality of UCRL but it does offer new theoretical insights into the regret guarantees of MDP.
> > >
> > > > Other reviewers have pointed out that several strong assumptions have been made, and potentially unfavorable (exponential) scalability wrt window sizes.
> > >
> > > Regarding Exponential Dependency on $k$ and $\ell$
> > >
> > > As we mentioned in the rebuttal to the other reviews, the exponential dependency in action is not avoidable in the worst case. Propositions 1 and 2 in [1] show that without extra assumptions, the sample complexity of any algorithm for POMDPs is $\Omega (A^{H-1})$. Our result is valuable since we identify a tractable class of POMDPs. Recall that our assumption requires the pseudo-invertibility of a matrix $M_h^\theta$ defined in Assumption 3.5. Intuitively, $M_h^\theta$ can be viewed as the sum of $O^kA^k$ rank-one matrices, which can be regular when $O^kA^k\geq d$. Similarly, Assumption 5.2 can be true when $A^\ell O^{\ell+1}\geq d$. Intuitively, the scale of $k$ and $\ell$ should be $\log d$ in our setting, under which the sample complexity of scale $O^{k+1}A^k$ is acceptable.
> > >
> > >
> > > [1] Jin, Chi, et al. "Sample-efficient reinforcement learning of undercomplete POMDPs." Advances in Neural Information Processing Systems 33 (2020): 18530-18539.
> > >
> > > [2] Liu, Qinghua, et al. "When Is Partially Observable Reinforcement Learning Not Scary?." arXiv preprint arXiv:2204.08967 (2022).

---

> > > > ### Comment · Reviewer_nfDQ · 2022-11-30
> > > > **Score revision**
> > > >
> > > > Alright, if everybody else thinks the paper should be accepted, I will not disagree. I have revised my score upwards to 6.

---

> > > > > ### Author Response · Authors · 2022-12-01
> > > > > **Response**
> > > > >
> > > > > We are glad that the reviewer's concerns have been addressed. Thanks for the valuable feedback and positive assessment of our work.

---

### Official Review · Reviewer_iXuJ · 2022-10-24

**Confidence:** 2
**Correctness:** 4
**Technical Novelty And Significance:** 2
**Empirical Novelty And Significance:** Not applicable
**Recommendation:** 6

**Clarity, Quality, Novelty And Reproducibility:**

The paper contains new results in the sense that they obtained an RL for POMDP with sample complexity.
The paper seems to have low readability due to notational complexities.

**Strength And Weaknesses:**

The strength of the paper is it obtained an RL algorithm for POMDPs, and obtained the corresponding sample complexity.
Weaknesses: The paper introduces several assumptions, and these assumptions seem to strong. Therefore, some discussions would be helpful.
Moreover, sufficient discussions on comparison with existing results are also lacking.

**Summary Of The Paper:**

The authors propose a reinforcement learning algorithm named Embed to Control (ETC), which learns the representation at two levels while optimizing the policy.
(i) For each step, ETC learns to represent the state with a low-dimensional feature, which factorizes the transition kernel. (ii) Across multiple steps, ETC learns to represent the full history with a low-dimensional embedding, which assembles the per-step feature. Using the two approaches, they developed RL algorithms that is scalable, and the corresponding sample complexities are obtained.

**Summary Of The Review:**

The paper seems to contain useful results.
The authors adopted several assumptions that are not clear to figure out how strong they are.
Therefore, it would be better to discuss more about the assumptions.
Some efforts are needed to improve readability of the paper.

---

> ### Author Response · Authors · 2022-11-17
> **Author response**
>
> Thank you very much for your valuable comments. We address the concerns as follows.
>
> >The authors adopted several assumptions that are not clear to figure out how strong they are. Therefore, it would be better to discuss more about the assumptions. Some efforts are needed to improve readability of the paper.
>
> Clarification of assumptions:
>
> Our assumptions are imposed for identifying the Bellman operator for POMDP (which is an extension of the Bellman operator in MDP). In particular,
>
> - Assumption 3.5 is required to ensure that the left inverse of $U_h^θ$ exists. The existence of a left inverse of $U_h^θ$ ensures that the information about the latent state can be extracted from future observations, which is also imposed by existing works on tabular POMDP [1][2].
>
> - More importantly, without such an assumption, there exists negative results showing that learning POMDP requires $\Omega(A^H)$ samples [3].
>
> - Our assumption that a left inverse of $U_h^θ$ exists is a natural extension of the assumptions made in the tabular case to the low-rank MDP setting.
>
> - Meanwhile, Assumption 5.2 specifies that the past observations contain enough information about the latent factors. Such an assumption is imposed to ensure that the model estimation aligns with policy evaluation.
>
> Note that Assumptions 3.5, 5.1, and 5.2 intrinsically only assume that $d$ vectors of dimensions $A^kO^{k+1}$ (or $A^\ell O^{\ell+1}$) are linearly independent, which is reasonable for relatively large action and observation spaces.
>
> >Therefore, some discussions would be helpful. Moreover, sufficient discussions on comparison with existing results are also lacking.
>
> Regarding related work:
>
> In our work, we compared our analysis with several concurrent and recent analyses of POMDP (e.g., [1][2][3]). See the Related Work in our submission for details. As discussed with Reviewer reeX, we will reorganize our paper and add more details in the comparison between our work and the related work. Please also let us know if there are other works that are relevant to our work.
>
> [1] Liu, Qinghua, et al. "When Is Partially Observable Reinforcement Learning Not Scary?." arXiv preprint arXiv:2204.08967 (2022).
>
> [2] Jin, Chi, et al. "Sample-efficient reinforcement learning of undercomplete POMDPs." Advances in Neural Information Processing Systems 33 (2020): 18530-18539.
> [3] Efroni, Yonathan, et al. "Provable reinforcement learning with a short-term memory." arXiv preprint arXiv:2202.03983 (2022).

---

> > ### Comment · Reviewer_iXuJ · 2022-12-14
> > **Response to the author's response**
> >
> > The authors addressed my previous comments well. I would like to keep my previous score.

---

### Official Review · Reviewer_6Ax4 · 2022-10-24

**Confidence:** 3
**Correctness:** 4
**Technical Novelty And Significance:** 2
**Empirical Novelty And Significance:** Not applicable
**Recommendation:** 6

**Clarity, Quality, Novelty And Reproducibility:**

Both the paper's main body and it's supplementary are well organized and clearly written.

The proposed theory and the ETC algorithmic appear to be novel.
Yet, this result should be taken with a grain of salt, as it involves:
i) strong assumptions; and
ii) the ETC algorithm has prohibitively large (exponential) sample complexity.

Minor Comments:

page 2: L^p space -> L^p norm space

page 4: rephrase -- "a embedding that are"

page 4: $\phi_h^{\theta}$ has domain $\mathcal{S} \times \mathcal{A}$

page 5: after "In addition, recall that we define": $\tau_h^k$ should be $\tau_h^{h+k}$

page 8: Alg. 1. -- Notation with left super-script for observations is not defined


**Strength And Weaknesses:**

Strengths

Makes a step towards building the foundations of a formal framework for analyzing POMDPs
under the assumptions that the transition kernel admits a low-rank decomposition with
i) past and future sufficiency; and ii) bounded norms of certain pseudo-inverses.


Weaknesses

1. There are several strong assumptions, see below. Further, *no argument is given to show
the existence of a range of parameters for which these assumptions are mutually satisfied*.

Assumption 3.5 (Future Sufficiency)
 - there exists $k>0$ such that $M$ is invertible

Assumption 5.1 (Bounded Pseudo-Inverse)
 - the norm of pseudo-inverse of $U$ is bounded ($\nu$)

Assumption 5.2 (Past Sufficiency)
 - there exists $\ell>0$ such that $F$ is left invertible
 - the norm of pseudo-inverse of $F$ is bounded ($\gamma$)

2. Under these assumptions, the ETC's algorithm scales *exponentially* with
the past and future window size. Namely, the term $|A|^{ O(k + \ell) }$ in Thm 5.3.

The ETC's sample complexity is prohibitively expensive, unless both $k$ and $\ell$ are either
i) constants independent of the horizon $H$; or
ii) are bounded by a slowly growing function of the horizon $H$.

**Summary Of The Paper:**

This work identifies four structural assumptions of low-rank POMDPs that allow for designing algorithms
with sample complexity that scales:
i) polynomially in the intrinsic dimension $d$ (transition kernel's rank) and the horizon length $H$; and
ii) *exponentially* with the past and future window sizes $\ell$ and $k$, respectively.

In essence, these assumptions ensure that the belief state can be recovered,
given past and future observation windows of sufficient length.

The proposed theory and algorithmic result (ETC) appear to be novel.
Yet, this work should be taken with a grain of salt (see Weaknesses), as it involves:
i) several strong assumptions; and
ii) the ETC algorithm has prohibitively large (exponential) sample complexity.

No empirical evidence is provided.

**Summary Of The Review:**

The paper is well written and contributes to the better tackling of POMDPs. However, it remains unclear to me to which extent this paper is a big step forward. The theoretical results are missing the set of parameters where all assumptions hold, and no empirical evidence is provided.

-- post rebuttal update --
Thanks for the clarifications. I increased my score to 6.

---

> ### Author Response · Authors · 2022-11-17
> **Author response**
>
> Thank you very much for your valuable comments. We address the concerns as follows.
>
> >There are several strong assumptions, see below. Further, no argument is given to show the existence of a range of parameters for which these assumptions are mutually satisfied.
>
> Regarding our Assumptions:
>
> Recall that our assumption requires the pseudo-invertibility of a matrix $M_h^{\theta}$ defined in Assumption 3.5. Intuitively, $M_h^{\theta}$ can be viewed as the sum of $O^kA^k$ rank-one matrices, which can be regular when $O^kA^k >d$. Similarly, Assumption 5.2 can be true when $ A^\ell O^{\ell+1}\geq d$. Intuitively, the scale of $k$ and $\ell$ should be $\log d$ in our setting, under which the sample complexity of scale $ O^{k+1}A^k $ is acceptable.
>
> In addition, note that Assumptions 3.5, 5.1, and 5.2 intrinsically only assume that $d$ vectors of dimensions $A^kO^{k+1}$ (or $A^\ell O^{\ell+1}$) are linearly independent, which is not a strong assumption when the action and observation spaces are large.
>
> >The ETC's sample complexity is prohibitively expensive, unless both $k$ and $\ell$ are either i) constants independent of the horizon $H$; or ii) are bounded by a slowly growing function of the horizon $H$.
>
> Regarding the Sample Complexity:
>
> As discussed in the comment before, we anticipate the scale of $k$ and $\ell$ to be at the $\log d$ scale in our setting. Thus, having $O^{k+1}A^k $ as the sample complexity is acceptable. In our analysis, the scale of $k$ and $h$ captures the hardness of solving the underlying POMDP. If we encounter the worst case where the observations have little information such that our assumptions do not hold even for large $k$ and $l$, there is little we can do as reinforcement learning in such a POMDP is near impossible. Indeed, there exist hardness examples [1] in such worst cases where you need $\exp(H)$ data points to learn the best policy.
>
> >Minor Comments:
>
> Thanks for the typo spotted. The left superscript denotes the data collected in the $t$-th iteration. See Appendix C for the details. We will revise and clarify accordingly.
>
> [1] Jin, Chi, et al. "Sample-efficient reinforcement learning of undercomplete POMDPs." Advances in Neural Information Processing Systems 33 (2020): 18530-18539.

---

> > ### Comment · Reviewer_6Ax4 · 2022-11-28
> > **Response to Authors**
> >
> > Thank you for your answers.
> >
> > Your argument gives a necessary, but not sufficient, condition for matrix $M_h^{\theta}(k)$ to be invertible. Due to the exponential dependence $A^{k}$, in order to claim a polynomial sample complexity on $d$, you have to either \emph{show} or \emph{assume} that $k \geq \log_{A} d^{O(1)} $ implies matrix $M_h^{\theta}(k)$ is invertible. Although such an assumption might be reasonable, it should be clearly stated.
> >
> > Taking into account other reviewers comments and your replies, I have decided to increase my score to 6.

---

> > > ### Author Response · Authors · 2022-12-01
> > > **Response**
> > >
> > > We are glad that the reviewer's concerns have been addressed. We will revise accordingly. Thanks for the valuable feedback and positive assessment of our work.

---

### Official Review · Reviewer_reeX · 2022-10-24

**Confidence:** 4
**Correctness:** 4
**Technical Novelty And Significance:** 3
**Empirical Novelty And Significance:** Not applicable
**Recommendation:** 6

**Clarity, Quality, Novelty And Reproducibility:**

Overall the paper is quite clearly presented, and the results are correct to my best effort of inspection. More detailed comments can be found in the above "strength and weaknesses" section.

**Strength And Weaknesses:**

Strengths:
* POMDPs is a challenging problem that received a lot of recent attention in the RL theory community, where additional tractability assumptions are required in order to achieve sample-efficient learning even in the basic tabular setting. The paper generalizes such studies into the infinite-state and observation setting and provides a set of sufficient conditions and a sample-efficient algorithm. This could serve as a starting point of future studies on the same topic.

* The past- and future-sufficiency conditions are tailored to the continuous setting, and are new to my best knowledge. The future-sufficiency condition is similar as the multi-step revealing condition of (Liu et al. 2022), but is slightly different in the choice of the norm ($1\to 1$ instead of $2\to 2$) on the matrix pseudo-inverse, which in my opinion is a better choice for the continuous setting (where $L_1$ norms have probabilistic meanings whereas $L_2$ norms may blow up).

* The algorithm achieves sample complexity that is polynomial in $A^{k+l}$ and all other problem parameters including the feature dimension and the past- and future-sufficiency parameters.

* The problem setting allows unknown features, generalizing the recent work of Cai et al. (2022) which assumes known features. This improvement is more minor though as now it is pretty well-understood how to do this in a modular fashion, e.g. in fully-observed MDPs by any algorithm for the Bellman rank or FLAMBE setting. Further, everything about function class is wrapped into the density estimation oracle in this paper.

Weaknesses:

* Despite the paper considers the continuous setting, it feels like the actual algorithm builds upon many existing algorithms, which does not seem to be adequately discussed in the main text. For example, the overall algorithm using confidence set (4.2) sounds like a global optimism algorithm similar as the OMLE algorithm of (Liu et al. 2022), where the risk functional (4.1) looks like the OOM-UCB type per-step risk function of (Jin et al. 2020a). It may be good to expand on these similarities in the paper, and what are the differences in the algorithm / analysis required in the current setting. In turn, maybe some current discussions like the “bottleneck factor interpretation” can be put into the appendix (it feels secondary, though I do like that discussion).

* Could the authors provide more intuitions on why the past-sufficiency parameter? The past-sufficiency looks a bit like the decodability condition (Efroni et al. 2022) for me, whereas future-sufficiency looks like the revealing condition. In tabular POMDPs, either one itself is sufficient for sample-efficient learning. Why does the present approach require both past- and future-sufficiency? Is it an artifact of the algorithm (in which case, pointing out why that is the case would be good for future understandings of this approach).

* Minor point: In the sample complexity, capacity of the function class is wrapped entirely in the density estimation coefficient $w_{\mathfrak{C}}$. For example, when $|\Theta|$ is finite, is it the case that we can always get $w_{\mathfrak{C}}\le \log|\Theta|$ automatically by MLE?


**Summary Of The Paper:**

This paper studies sample-efficient learning of POMDPs. The paper considers POMDPs with infinite (continuous) state and observation spaces and admitting low-rank latent transitions with features belonging to a certain feature set. The main result is the identification of a sufficient condition called past- and future-sufficiency, along with an algorithm Embed-To-Control (ETC) that achieves sample-efficient learning in this setting.

**Summary Of The Review:**

The paper provides results for sample-efficient learning of POMDPs with continuous state/observation spaces with unknown features, though it may be a more direct extension of recent works along this line than currently claimed.

---

> ### Author Response · Authors · 2022-11-17
> **Author response**
>
> Thank you very much for your valuable comments. We address the concerns as follows.
>
> > Despite the paper considers the continuous setting, it feels like the actual algorithm builds upon many existing algorithms, which does not seem to be adequately discussed in the main text. For example, the overall algorithm using confidence set (4.2) sounds like a global optimism algorithm similar as the OMLE algorithm of (Liu et al. 2022), where the risk functional (4.1) looks like the OOM-UCB type per-step risk function of (Jin et al. 2020a). It may be good to expand on these similarities in the paper, and what are the differences in the algorithm / analysis required in the current setting. In turn, maybe some current discussions like the “bottleneck factor interpretation” can be put into the appendix (it feels secondary, though I do like that discussion).
>
> Regarding our Algorithm:
>
> Thanks for the additional reference recommended. Indeed, our work is inspired by the previous OOM-UCB analysis. [1] is also a concurrent work that extends the OOM-UCB type analysis to multi-steps. In contrast,  Liu et al. 2022 consider the finite state and observation spaces, whereas we consider the infinite state space cases, which are inherently different. In particular, their technique cannot directly be adopted to the settings that we consider, as will be discussed below in detail. In our revision, we will organize the structure of our paper and put the bottleneck factor interpretation in the appendix to make some space for the discussion of concurrent works.
>
> >Could the authors provide more intuitions on why the past-sufficiency parameter? The past-sufficiency looks a bit like the decodability condition (Efroni et al. 2022) for me, whereas future-sufficiency looks like the revealing condition. In tabular POMDPs, either one itself is sufficient for sample-efficient learning. Why does the present approach require both past- and future-sufficiency? Is it an artifact of the algorithm (in which case, pointing out why that is the case would be good for future understandings of this approach).
>
> Regarding our Future-Sufficiency Assumption:
>
> Our assumption is different from the decodability assumption by [3]. In particular, the decodability assumption ensures that the exact latent state can be inferred by the history interactions. In other words, the latent state is a deterministic function of the history of interactions. In contrast, in our setting, we allow random latent states given the interaction history, that is, even when the history is fixed, only the distribution of the latent state is fixed, but the latent state itself is still a random variable.
>
>  Although the future sufficiency condition seems similar to the revealing condition in the previous analysis, it is a novel assumption in the sense that we generalize the revealing condition to continuous state and action spaces.
>
> Regarding our Past-Sufficiency Assumption:
>
> We introduce the past sufficiency condition to handle the difficulty in sample efficiency analysis. Here we want to use the interaction history to infer the next-step transition. Note that the transition can be seen as an infinite-by-infinite matrix, which is impossible to infer without extra structural assumptions. By introducing the future and past sufficiency conditions, we can represent such an infinite-by-infinite matrix using finite-dimensional representations, thus allowing sample-efficient estimation. A variant of such an assumption is also imposed by recent work [2].
>
> We highlight that directly applying the technique for tabular analysis cannot handle the setting with continuous state and action spaces.
> In particular, the tabular-based analysis results in a logarithmic factor in state spaces (e.g., [1]),
> such logarithmic factor becomes infinite in continuous state spaces.
> Our analysis shows that utilizing an additional past sufficiency assumption is sufficient for the sample efficiency analysis.
>
> >Minor point: In the sample complexity, capacity of the function class is wrapped entirely in the density estimation coefficient . For example, when  is finite, is it the case that we can always get  automatically by MLE?
>
> Regarding the MLE:
>
> Yes indeed. For the finite function class case the coefficient is upper bounded by the logarithmic factor of function capacity.
>
> [1] Liu, Qinghua, et al. "When Is Partially Observable Reinforcement Learning Not Scary?." arXiv preprint arXiv:2204.08967 (2022).
>
> [2] Zhan, Wenhao, et al. "Pac reinforcement learning for predictive state representations." arXiv preprint arXiv:2207.05738 (2022).
>
> [3] Efroni, Yonathan, et al. "Provable reinforcement learning with a short-term memory." arXiv preprint arXiv:2202.03983 (2022).

---

### Decision · Program_Chairs · 2023-01-20

**Decision:**

Accept: poster

**Justification For Why Not Higher Score:**

A thorough discussion of the assumptions, their relation to those used in the prior work, and how restricting they are is missing. Moreover, there is no experimental results to support the theoretical findings.

**Justification For Why Not Lower Score:**

The paper is relatively well-written. It provides non-trivial and novel theoretical and algorithmic results POMDPs. It identifies four structural assumptions of low-rank POMDPs that allow for designing polynomial (in intrinsic dimension and horizon) algorithms.

**Metareview: Summary, Strengths And Weaknesses:**

The reviewers see the paper as a step towards building a framework for analyzing POMDPs. They found its writing reasonable and the theoretical and algorithmic results satisfactory. However, there are concerns about the assumptions: how strong they are, how they are related to the assumptions used in the prior work, etc. Moreover, the lack of empirical evaluation seems to be an issue (rather a minor issue). I would suggest that the authors take these two points, especially the first one, into consideration and add a thorough discussion of the assumptions in the next revision of their paper.

**Note From Pc:**

if the above contains the word "oral" or "spotlight" please see: "oral" presentation means -> notable-top-5% and "spotlight" means -> notable-top-25%. As stated in our emails, we are disassociating presentation type from AC recommendations